# CONTRASTIVE SELF-REWARDING MLLM

## ABSTRACT

Direct Preference Optimization (DPO) has been proven effective and efficient in aligning Multi-modal Large Language Models (MLLMs) with human preferences and improving multimodal understanding of MLLMs. However, most existing work relies heavily on either human annotations or auxiliary reward models to construct preference data, which limits their scalability and introduces potential inconsistencies between the reward model and fine-tuned MLLMs. This paper presents ConSR, a Contrastive Self-rewarded Preference Optimization framework that constructs contrastive inputs and frames the variation of the corresponding model outputs as self-reward signals. We perturb the visual input by degrading its fine-grained details and enriching it with semantic context, respectively, forming contrasts with the original visual input. The variation of the corresponding model responses reveals the model's sensitivity to the visual inputs, which we exploit to rank and construct preference pairs without external supervision. In addition, we reformulate the DPO objective to mitigate its length bias, and reweight visual tokens to assign higher weights to more responsive tokens regarding visual cues. Extensive experiments across multiple visual understanding benchmarks demonstrate ConSR's consistent superiority across diverse tasks.

## 1 INTRODUCTION

Multi-modal Large Language Models (MLLMs) (Wang et al., 2024c; Liu et al., 2024a; Chen et al., 2024g; Bavishi et al., 2023; Hu et al., 2024; Bai et al., 2023; Zhu et al., 2023; Dai et al., 2024) have achieved significant progress in image-language understanding and reasoning. Despite these advances, their responses often do not align with human preferences. Several recent studies address this issue by introducing Direct Preference Optimization (DPO) (Rafailov et al., 2023) that fine-tunes MLLMs by using pairwise comparisons between preferred and dispreferred responses. However, they typically rely on costly human annotations or external advanced MLLMs (e.g., closed-source GPT-4) to generate synthetic preference data, both introducing substantial cost and degrading scalability.

Several approaches have been explored to alleviate the need of manual supervision by constructing preference data without human labels or external models. One representative approach repurposes the MLLM itself as both the policy and the reward model, prompting it with "LLM-as-a-Judge" instructions (Yu et al., 2024b; Zheng et al., 2023; Wang et al., 2024f;e) to rank its own outputs. However, this approach assumes that the model has strong judging capabilities—a property that usually requires additional fine-tuning and is not guaranteed in general-purpose MLLMs. Another approach (Zhou et al., 2024a; Zhu et al., 2024; Chen et al., 2025) generates high-contrast preference pairs by corrupting the visual input (e.g., injecting noises into the image) to induce degraded responses and uses them as dispreferred examples. However, it primarily targets on mitigating hallucination rather than enhancing the general visual understanding capability of MLLMs.

To address these limitations, we explore the use of self-contrastive signals to estimate rewards, without relying on human annotations or LLM-based judges. Such signals can be derived by measuring how a model's output distribution shifts in response to controlled input perturbations. The core intuition is that high-quality responses should be more sensitive or supported by meaningful enhancement of the input context, while low-quality responses remain less affected. This contrastive behavior, much like how humans become more confident in answering questions or reasoning as they receive additional cues, could serve as a form of intrinsic rewards for preference optimization. By encouraging the model to favor such contrast-sensitive responses, we facilitate a stronger reliance of MLLMs on visual input, even in the absence of external supervision.

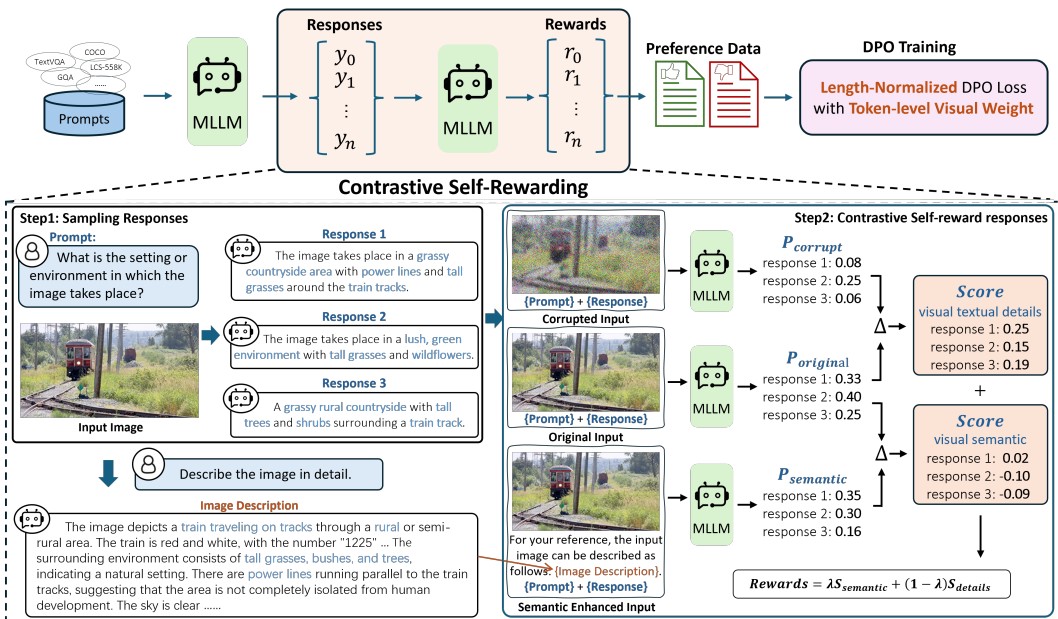

Figure 1: The framework of the proposed Contrastive Self-Rewarding (ConSR). Given an input prompt and image, ConSR first samples multiple candidate responses from a multi-modal large language model (MLLM). To estimate visual dependency, it evaluates the response probabilities under three conditions: (1) the original image, (2) a corrupted image with degraded visual details (via diffusion noise), and (3) an enhanced input with a model-generated image description appended to the prompt. Based on the probability shifts, we compute contrastive self-reward scores reflecting the response's sensitivity to semantic and detailed visual cues. These scores are used to rank responses and construct preference pairs. We also propose a length-normalized DPO loss with token-level visual weighting to further enhance the image preference of MLLMs.

In this work, we propose Contrastive Self-Rewarding (ConSR), a novel self-rewarding approach that constructs preference data by employing probability shifts of model responses as reward signals. As illustrated in Figure 1, ConSR creates two input variants that either **enhance** or **corrupt** original visual signals to enable contrastive inputs. For **corrupt**, we apply diffusion-based Gaussian noise to images, which degrades local textures while preserving overall structure. For **enhance**, we augment original image inputs with detailed image descriptions generated by MLLM itself to enrich the context in question answering. After sampling multiple responses from target MLLMs, we evaluate how their output probabilities shift under each perturbed input and ensemble such shifts from both corruption and enhancement to compute a self-reward score, which favors responses whose probability increases in the presence of more informative visual signals. This enables an automated construction of preference data that is grounded in both semantic content and textual detail of visual inputs.

Beyond preference data construction, we also reformulate the DPO loss function by migating its inherent bias toward longer responses (Park et al., 2024; Lu et al., 2024; Chen et al., 2024a) and further incorporating token-level training signals. Unlike prior works that modify the DPO objective directly, we first decompose the loss into a more interpretable form consisting of a scaling weight and standard cross-entropy terms for positive and negative samples, derived from the gradient expression of the original DPO objective. To reduce the length bias, we normalize the weighting coefficient in DPO loss based on the length of each response. Additionally, similar to our proposed contrastive rewards, we introduce token-level visual weights, assigning higher weight to tokens whose output probabilities increase with enhanced input. The newly formulated DPO objective and our contrastive self-reward data are complementary which together improve performance consistently across base models and a wide range of visual understanding and hallucination benchmarks.

The contributions of this work can be summarized in three major aspects.

- We propose ConSR, a contrastive self-rewarding approach, to leverage the probability shift under contrastive visual inputs as reward signals to curate preference data. We measure how each data responds to the enhancement or corruption of the input context to select the preference data that best aligns with the input context.

- We incorporate a novel length-normalized DPO loss with token-level visual weight for fine-grained preference supervision.

- Empirical results demonstrate that our methods achieve consistent improvements on two strong MLLM backbones, LLaVA-OneVision and Qwen2.5-VL, across a diverse set of image understanding and hallucination benchmarks.

## 2    RELATED WORKS

**Multimodal Large Language Models**. The recent emergence of Large Language Models (LLMs) has facilitated the development of their multimodal counterparts by integrating visual features, leading to the thrive of Multimodal Large Language Models (MLLMs) (Li et al., 2025) In recent years persistent efforts have been made in this field to expand MLLMs skill sets, from simple perception tasks (Fu et al., 2023; Liu et al., 2023c; Nie et al., 2024) to complex reasoning scenarios (Lu et al., 2023; Wang et al., 2024b; Yang et al., 2024; Hu et al., 2025), and from handling single image (Liu et al., 2023a; 2024b) to hour-long video inputs (Shu et al., 2024; Chen et al., 2024e). Despite the achievements in open-source MLLMs, these models still rely on carefully collected, high-quality instruction data (Chen et al., 2024b;d) for post-training. We explore how to enable self-reward without explicit supervision.

**Learning from Feedback**. Guiding LLMs and MLLMs to learn from feedback has facilitated the emergence of strong power in these models. Proximal Policy Optimization (PPO) (Schulman et al., 2017) and Direct Preference Optimization (DPO) (Rafailov et al., 2023) are the recent de-facto approaches for aligning LLMs with human preferences. The success of DPO in LLMs has been replicated in their multimodal counterparts recently. Early studies that apply such technique to MLLMs rely on human labeled preferences (Sun et al., 2023; Yu et al., 2024a). Despite its success in addressing issues such as hallucinations (Bai et al., 2024; Yifan et al., 2023), it relies on resource-intensive data collection, which hinders its scalability. To alleviate the labeling burden, a line of recent studies explore using AI judges to provide feedback with external models (Li et al., 2023; Xiao et al., 2025; Jing & Du, 2024; Pi et al., 2024; Yu et al., 2024b; Zhao et al., 2023; Zhou et al., 2024a; Zhang et al., 2024b), such as GPT-4V (Yu et al., 2024b), Stable Diffusion (Xie et al., 2024), or customized models (Xiong et al., 2024; Zhang et al., 2024a). The external judges may bring additional bias, making the estimated preferences less effective. Distinctively, our approach explores how MLLMs can improve by themselves without relying on external feedbacks and how self-contrastive signals serve as rewards for DPO training.

**Self-improve for MLLMs**. Self-improvement of LLMs (Huang et al., 2022; Chen et al., 2024h; Yuan et al., 2025; Wang et al., 2025; Kumar et al., 2025) has been an intriguing feature for these models recently, as it suggests potential to improve LLMs by themselves without explicit human supervision. With only observing input sequences, pretrained LLMs can improve themselves via finetuning on high-confidence predictions selected by majority voting (Huang et al., 2022; Zuo et al., 2025; Wang et al., 2022). Alternatively, (Chen et al., 2024h) implements self-play of LLMs with the objective of discrimination between model-generated and human-annotated data. Few studies explore this problem in the scope of multi-modalities (Zhou et al., 2024b; Wang et al., 2024d; Chen et al., 2025), where leverage MLLMs themselves to construct preference data. We present that self-contrastive signals can also serve as self-reward signals that enhance MLLM visual understanding.

## 3    METHOD

In this work, we propose a contrastive self-reward approach with the reformulated DPO loss for MLLM preference optimization. Our approach consists of three key steps: (1) response sampling from MLLMs in Section 3.1, (2) contrastive self-reward computation based on input perturbations in Section 3.2, and (3) length-normalized DPO loss with token-level visual weighting in Section 3.3.

## 3.1 RESPONSE SAMPLING

For preference optimization, we start by sampling a set of $K$ candidate responses from the policy model $\pi_\theta$, given an input image $I$ and a multi-modal prompt $x$ (e.g., a visual question). We set the temperature to 1.2 to encourage response diversity and use $K = 5$ by default. To further enhance the diversity and robustness of the candidate pool, we introduce two additional response strategies: response revision and noisy sampling.

Response revision is motivated by our observation that MLLMs may fail to generate correct or desired answers even with multiple samples. To reduce the risk of learning from preference data with error, we prompt the model to revise its initial response using a self-generated image description as additional context. During revision, the model is instructed to retain the original response's structure and style as much as possible. The specific prompt used for revision is detailed in the Appendix.

Inspired by prior work (Zhou et al., 2024a), we adopt noisy sampling to enhance the diversity of model responses. Specifically, we apply diffusion-based noise (Ho et al., 2020) to the input image and sample additional responses from the resulting corrupted inputs. The noise level is controlled using 500 diffusion steps, which introduces moderate degradation while preserving the overall semantics of the image. This setup encourages the MLLM to generate slightly altered, yet still meaningful responses that remain close to its original response distribution.

In total, we collect 15 candidate responses per input query from the model with the combination of direct sampling, revision, and noise-based sampling. These responses are then scored and ranked using our proposed contrastive self-reward to construct pairwise data for preference optimization.

## 3.2 CONTRASTIVE SELF-REWARDING

Our proposed Contrastive Self-Rewarding (ConSR) leverages diverse sampled response pairs to construct vision-centric preferences without relying on explicit human supervision. The intuition is that responses that are highly sensitive to input perturbations are more likely vision-centric, while responses that are less affected by such perturbations might be dominated by statistical bias or language priors. Our approach shares its underlying motivation with recent training-free contrastive decoding methods (Leng et al., 2023; Neo & Chen, 2024; Chen et al., 2024f), which evaluate response quality based on changes in model confidence under perturbed inputs. However, it remains underexplored how such contrastive signals can be utilized in the post-training stage of MLLMs, and whether they can benefit a broader range of vision-language tasks beyond hallucination mitigation.

To comprehensively evaluate how well each candidate's response aligns with the visual input, we decompose the image into two components: global semantics and fine-grained texture details. We then perturb each component separately to observe how the model's output probability shifts, thereby ranking the responses. The overall process is illustrated in Figure 1. Given an image $I$, a prompt $x$, and a sampled response $y$, we first compute the model's generation probability for the response using the average token-level log-probability:

$$P_\theta(y \mid x, I) = \exp\left(\frac{1}{T}\sum_{t=1}^{T}\log \pi_\theta(y_t \mid y_{<t}, x, I)\right) \tag{1}$$

To perturb fine-grained visual details, we apply Gaussian diffusion noise to the original image, resulting in a corrupted version $\tilde{I}$. As shown in Figure 1, after adding 800 steps of diffusion noise to the image, the fine-grained texture details are heavily corrupted. However, the overall image content, such as the tracks, grass, and trains, still remains vaguely recognizable. The model's response probability under corrupted input is computed as:

$$P_\theta^{corrupt}(y \mid x, \tilde{I}) = \exp\left(\frac{1}{T}\sum_{t=1}^{T}\log \pi_\theta(y_t \mid y_{<t}, x, \tilde{I})\right) \tag{2}$$

For visual semantics, we note that it is difficult to directly corrupt high-level semantic information in an image. Therefore, instead of corrupting visual semantics, we enhance the model's semantic context in answering questions by modifying the prompt. Specifically, we prompt the model to generate a

detailed image description $d$, and insert this description into the original input to form an enriched prompt $x' = d \oplus x$. The model's response probability under enriched semantics is then computed as:

$$P_\theta^{semantic}(y \mid x', I) = \exp\left(\frac{1}{T}\sum_{t=1}^{T} \log \pi_\theta(y_t \mid y_{<t}, x', I)\right) \tag{3}$$

We then leverage the probability shifts under these two modifications as responses' scores for visual text details and semantics. The final contrastive self-reward score for response $y$ is then computed as a weighted combination of both scores:

$$S_{\text{semantic}}(y) = P_\theta^{\text{semantic}}(y \mid x', I) - P_\theta(y \mid x, I), \tag{4}$$

$$S_{\text{corrupt}}(y) = P_\theta(y \mid x, I) - P_\theta^{\text{corrupt}}(y \mid x, \tilde{I}), \tag{5}$$

$$r(y) = \lambda \cdot S_{\text{semantic}}(y) + (1 - \lambda) \cdot S_{\text{corrupt}}(y). \tag{6}$$

Here, $\lambda$ balances the importance between semantic sensitivity and fine-grained visual dependency. Responses with the highest $r(y)$ are considered more aligned with the visual input and selected as preferred data. The constructed preference data are then adopted for preference optimization.

### 3.3 LENGTH-NORMALIZED DPO LOSS WITH TOKEN-LEVEL VISUAL WEIGHTS

DPO is a promising approach for preference alignment compared to other approaches due to its simplicity and training efficiency. However, DPO exhibits two limitations in the multimodal setting: (1) length bias (or length exploitation), where preference alignment approaches may prefer a longer response and lead optimized models to generate more verbose or longer responses (Park et al., 2024; Lu et al., 2024), and (2) lack of fine-grained visual supervision, as standard DPO applies the same learning signal uniformly across all tokens. We thus make two modifications to the typical DPO loss, including length normalization and token-level visual weights, which are elaborated on below.

To address these issues, we first reformulate the DPO objective into a weighted cross-entropy form that clearly separates the loss into interpretable components. The DPO adopts the following objective:

$$\mathcal{L}_{\text{DPO}}(\pi_\theta; \pi_{\text{ref}}) = -\mathbb{E}_{(x,y_w,y_l)\sim\mathcal{D}}[\log \sigma(\beta \log \frac{\pi_\theta(y_w \mid x, I)}{\pi_{\text{ref}}(y_w \mid x, I)} - \beta \log \frac{\pi_\theta(y_l \mid x, I)}{\pi_{\text{ref}}(y_l \mid x, I)})] \tag{7}$$

where $\pi_\theta$ and $\pi_{\text{ref}}$ denotes policy model and reference model, $y_w$ and $y_l$ indicates preferred and dispreferred responses. Its gradient with respect to the $\theta$ can be written as:

$$\nabla_\theta \mathcal{L}_{\text{DPO}}(\pi_\theta; \pi_{\text{ref}}) = -\beta \mathbb{E}_{(x,y_w,y_l)\sim\mathcal{D}}[\sigma(-\Delta)(\nabla_\theta \log \pi_\theta(y_w \mid x, I) - \nabla_\theta \log \pi_\theta(y_l \mid x, I))] \tag{8}$$

$$\Delta = \beta \log \frac{\pi_\theta(y_w \mid x, I)}{\pi_{\text{ref}}(y_w \mid x, I)} - \beta \log \frac{\pi_\theta(y_l \mid x, I)}{\pi_{\text{ref}}(y_l \mid x, I)} \tag{9}$$

where $\sigma(-\Delta)$ weights the gradient by how incorrectly implicit rewards order $y_l$ over $y_w$, and the remaining parts are gradients for the positive and negative cross-entropy losses of preference data.

**Length-normalized Loss.** Our analysis centers on how the length of preference data affects the weighting in the DPO loss formulation. The implicit reward in the weight can be written as an accumulation in token-level reward $\sum_{t=1}^{|y|} \log \frac{p_\theta(y_t \mid x, I, y_{<t})}{p_{\text{ref}}(y_{w,t} \mid x, I, y_{w,<t})}$, where longer sequences tend to accumulate larger value. Therefore, a longer dispreferred response might lead to higher weight. However, in practice, during DPO training, both $y_w$ and $y_l$ typically yield negative reward values (Ren & Sutherland, 2024). As a result, when the preferred response $y_w$ is longer than the dispreferred one $y_l$, the scaling weight becomes larger in magnitude, assigning greater weight to the gradients of cross-entropy losses.

To mitigate this effect, we first reformulate the loss into the following format, which has the same gradient as Equation 7:

$$\mathcal{L}_{\text{DPO}}(\pi_\theta; \pi_{\text{ref}}) = -\beta \mathbb{E}_{(x,y_w,y_l)\sim\mathcal{D}}[detach(\sigma(-\Delta))(\log \pi_\theta(y_w \mid x) - \log \pi_\theta(y_l \mid x))] \tag{10}$$

Then, to alleviate the length bias, we normalize the reward terms of $y_w$ and $y_l$ by their lengths, ensuring that the weights reflect the gap between responses' implicit rewards without any length bias. The length-normalized weight is computed as follows:

$$\Delta' = \frac{|y_w| + |y_l|}{2} \left( \frac{\beta}{|y_w|} \sum_{t=1}^{|y_w|} \log \frac{p_\theta(y_{w,t} \mid x, I, y_{w,<t})}{p_{\text{ref}}(y_{w,t} \mid x, I, y_{w,<t})} - \frac{\beta}{|y_l|} \sum_{t=1}^{|y_l|} \log \frac{p_\theta(y_{l,t} \mid x, I, y_{l,<t})}{p_{\text{ref}}(y_{l,t} \mid x, I, y_{l,<t})} \right) \quad (11)$$

**Token-level visual weight.** Another limitation of DPO lies on the lack of token-level supervision. To guide MLLMs with fine-grained supervision, we introduce token-level weights into the DPO loss based on each token's probability shifts, while more visual clues are provided. Specifically, for a given token in the response $y$, we first obtain its probability shift as:

$$\Delta p_y^t = p_\theta(y_t \mid x, I, y_{<t}) - p_\theta(y_t \mid x, \tilde{I}, y_{<t}) \quad (12)$$

Similar to our contrastive rewards in Section 3.2, we aim to reward those tokens with higher probabilities from the corrupted image to the original image. Therefore, we introduce a visual weight into our reformulated for each token as:

$$\omega_t = \begin{cases} 1 + \text{detach}(\text{clip}\left(\alpha \cdot \Delta p_y^t, 0, \varepsilon\right)), & \text{if } y \in y_w \\ 1 - \text{detach}(\text{clip}\left(\alpha \cdot \Delta p_y^t, 0, \varepsilon\right)), & \text{if } y \in y_l \end{cases} \quad (13)$$

where $\alpha$ controls the scale of weight and $\varepsilon$ set an upbound of weight. For each token in the preferred response $y_w$, we apply a final weight of $(1 + w)$, and for the dispreferred response $y_l$, we apply $(1 - w)$. These weights are detached from the gradient to prevent backpropagation through the weight computation itself. The final length-normalized loss with token visual weights is shown as follows:

$$\mathcal{L}' = -detach(\sigma(-\Delta')) \cdot \left[ \sum_{t=1}^{|y_w|} \omega_t \cdot \log \pi_\theta(y_{w,t} \mid x, I, y_{w,<t}) - \sum_{t=1}^{|y_l|} \omega_t \cdot \log \pi_\theta(y_{l,t} \mid x, I, y_{l,<t}) \right]$$
$$(14)$$

Several recent works (Yang et al., 2025; Wang et al., 2024a; Gu et al., 2024; Neo & Chen, 2024) share a similar motivation with ours: leveraging contrastive inputs to provide additional supervision to the DPO loss. For example, mDPO (Wang et al., 2024a) replaces the implicit rewards of $y_l$ in DPO with a reward of $y_w$ under a corrupted image, while TPO (Gu et al., 2024) scales the original DPO rewards using a token-level reward scheme, both aiming to enhance the model's preference for visual input. However, these methods directly incorporate the model's probability under corrupted images into the DPO loss, which leads the model to optimize on corrupted inputs directly. In contrast, our approach reformulates the DPO objective and introduces detached token-level weights to encourage visual preference without explicitly optimizing on noisy inputs. This formulation offers a more interpretable training signal while avoiding potential instability caused by learning from corrupted visual content.

## 4 EXPERIMENTS

### 4.1 IMPLEMENTATION DETAILS

**Training.** We adopt two advanced MLLMs, LLaVA-OneVision (LLaVA-OV) (Li et al., 2024a) and Qwen2.5-VL (Bai et al., 2025), as our base model, considering their superior performance, which facilitates sampling valuable responses to construct preference data. We sample 10k instructions from the RLAIF-V (Yu et al., 2024c) dataset to prompt MLLM to generate responses. The instructions are from various datasets such as OK-VQA (Marino et al., 2019), VQAv2, sharegpt4v (Chen et al., 2024b), LCS-558K (Liu et al., 2023a), etc. The diversity of prompts ensures a general use for improving MLLM's comprehensive capabilities. Both LLaVA-OV and Qwen2.5-VL are fine-tuned for one epoch on the self-rewarded preference data. The entire training process requires 7 and 5 hours

Table 1: Experiments on general image understanding benchmarks. The upper part shows results reported in the original papers, while the lower part contains our reproduced results (marked with †). Our method effectively improves the base models across seven benchmarks.

| Model | Seed2+ | OCRBench | MMMUPro | LLaVA | MMStar | MMB | MME |
|---|---|---|---|---|---|---|---|
| MiniCPM-V-2.6 | - | 85.2 | 30.2 | - | 57.5 | - | 2348 |
| LLaVA-OV-7B | - | 62.3 | - | 90.7 | 61.7 | 80.8 | 1998 |
| MM-RLHF | 65.4 | 69.3 | - | 97.9 | 62.6 | - | 2025 |
| InternVL2-8B | 67.5 | 79.4 | 32.5 | - | 62.0 | 81.7 | 2210 |
| InternVL2.5-8B | 69.7 | 82.2 | 38.2 | - | 62.8 | 84.6 | 2344 |
| Qwen2-VL-7B | 69.0 | 84.5 | 34.1 | - | 60.7 | 83.0 | 2326 |
| Qwen2.5-VL-7B | 70.4 | 86.4 | - | - | 63.9 | 83.5 | 2347 |
| LLaVA-OV-7B† | 64.8 | 62.1 | 29.8 | 90.7 | 61.7 | 80.8 | 1998 |
| + ConSR (ours) | 66.6 | 67.9 | 30.6 | 93.2 | 62.8 | 81.2 | 2001 |
| Qwen2.5-VL-7B† | 70.6 | 88.2 | 35.0 | 103.0 | 64.4 | 83.7 | 2333 |
| + ConSR (ours) | 71.1 | 88.6 | 36.1 | 106.9 | 65.5 | 84.1 | 2344 |

Table 2: Experiments on hallucination benchmarks. We evaluate all models under the same settings and environment (with †).

| Model | POPE(f1) | POPE(Acc) | MMHal | HallBench | AMBER |
|---|---|---|---|---|---|
| LLaVA-OV-7B† | 88.1 | 89.0 | 2.97 | 47.4 | 87.3 |
| + ConSR (ours) | 88.7 | 89.4 | 3.30 | 49.3 | 87.7 |
| Qwen2.5-VL-7B† | 86.3 | 87.6 | 3.73 | 53.7 | 85.1 |
| + ConSR (ours) | 86.5 | 87.8 | 3.74 | 54.2 | 85.4 |

for LLava-OV-7B and Qwen2.5-VL-7B on one A100 GPU. More training parameters are detailed in the Appendix.

**Evaluation Benchmarks.** We comprehensively evaluate our methods on both general image understanding benchmarks and hallucination benchmarks. For general image understanding, we evaluate on seven benchmarks, including Seedbench2Plus (Li et al., 2024b) and OCRBench for multimodal OCR and text-related abilities, LLaVA-Bench (Liu et al., 2023b) for multimodal conversation, MMMU_Pro (Yue et al., 2024) for multimodal reasoning, MMstar (Chen et al., 2024c),MMbench_EN (Liu et al., 2023c), and MME (Fu et al., 2023) for comprehensive multimodal capabilities. These benchmarks cover a variety of MLLM's capabilities for multimodal recognition, perception, and reasoning. We also evaluate the models on four hallucination benchmarks, POPE, Hallusion bench, MMHal, and AMBER. We adopt both LMMS-eval (Li* et al., 2024) and VLMEvalKit (Duan et al., 2024) to evaluate benchmarks to try to match the original reported performances of evaluated MLLMs. All the ablation experiments are conducted on LLaVA-OV-7B.

## 4.2 MAIN RESULTS

In Table 1, we report results on seven general image understanding benchmarks, comparing our method with advanced MLLMs. We first apply our approach to LLaVA-OV-7B, where it achieves consistent improvements across all benchmarks, with gains of +1.1 on MMStar, +1.8 on MMMUPro, and +2.5 on the LLaVA benchmark. Notably, we observe significant gains of +1.8 on SeedBench2+ and +5.8 on OCRBench, demonstrating substantial enhancement in multimodal OCR-related capabilities. To evaluate the generality of our method, we further apply ConSR to Qwen2.5-VL-7B, a model from a different architecture family with generally stronger baseline performance. Similar to LLaVA-OneVision, ConSR consistently improves Qwen2.5-VL-7B across all benchmarks. Interestingly, the improvements vary in focus: Qwen2.5-VL-7B achieves larger gains on MME (+11.0), reflecting stronger improvements in comprehensive multimodal reasoning, while LLaVA-OV-7B benefits more on OCR-related tasks. This difference may stem from their response styles, Qwen2.5-VL-7B tends to produce more detailed and structured answers, thus resulting in preference data that benefits more on comprehensive multimode capabilities. Overall, these results validate the effectiveness and

Table 3: Experiments on the construction of preference data. "$y_l$ from Noise Image" denotes using the original MLLM response and a response from a Gaussian-noised image as preferred and dispreferred samples, respectively. "All data" includes the combination of normally sampled, revised, and noisy responses. Results are reported on six benchmarks.

| Model | Seed2Plus | OCRBench | MMMUPro | LLaVA | MMStar | MMB |
|---|---|---|---|---|---|---|
| LLaVA-OV-7B | 64.8 | 62.1 | 29.8 | 90.7 | 61.7 | 80.8 |
| + $y_l$ from Noise Image | 65.7 | 64.5 | 30.0 | 92.1 | 62.1 | 80.3 |
| + Preference data ranked by ConSR ($\lambda$ in Equation 6) | | | | | | |
| All data $\lambda = 0.4$ | 66.5 | 67.0 | 30.9 | 88.6 | 62.2 | 81.2 |
| All data $\lambda = 0.6$ | 66.3 | 68.0 | 30.8 | 91.9 | 62.8 | 80.8 |
| All data $\lambda = 0.8$ (default) | 66.6 | 67.9 | 30.6 | 93.2 | 62.8 | 81.2 |
| All data $\lambda = 1.0$ | 65.5 | 65.8 | 30.2 | 93.4 | 62.0 | 80.0 |
| w/o revise & noise data | 65.4 | 61.8 | 30.5 | 86.7 | 63.0 | 81.2 |

generalization of our method in enhancing general image understanding without requiring additional supervision or architectural modifications.

To evaluate the impact of ConST on hallucination mitigation, we conduct experiments on four widely used benchmarks: POPE, MMHal, HallBench, and AMBER. As shown in Table 2, although ConSR is not specifically designed to mitigate hallucinations, it consistently improves performance across all benchmarks for both LLaVA-OV-7B and Qwen2.5-VL-7B. On POPE, ConSR improves F1 and accuracy by +0.6 and +0.4, respectively, for LLaVA. On more challenging hallucination evaluations such as MMHal, HallBench, and AMBER, ConSR achieves gains of +0.33, +2.8, and +0.4, respectively, indicating improved trustworthiness of MLLMs under complex scenarios. These results suggest our proposed ConSR brings consistent gains on both general image understanding and hallucination benchmarks by promoting better alignment between model responses and visual input.

### 4.3 ABLATION STUDY

To comprehensively evaluate our design, we experiment with the construction of preference data and the design of the loss function separately. As shown in Table 3, we investigate different strategies for generating contrastive responses and the effect of reward balancing between semantic and detail-level signals. Directly using corrupted responses like responses $y_l$ from noise images as dispreferred data have been explored by previous methods in mitigating hallucinations (Zhou et al., 2024a; Chen et al., 2025; Amirloo et al., 2024). However, this setting only yields modest gains on most general image understanding benchmarks, such as Seed2Plus (+0.9) and LLaVA (+1.4), and slightly underperforms on MMBench due to potential misalignment in overly corrupted responses. We further vary the weighting parameter $\lambda$ (Equation 6) to control the balance of the reward signal from semantic enrichment and detail corruption. The highest performance is adopted at $\lambda = 0.8$ (our default), suggesting that semantic and fine-grained cues are important for contrastive supervision. Lastly, removing revision and noise-based responses (w/o revise & noise data") slightly reduces the overall gains, validating the effect of diversity of candidate responses in improving preference data quality.

In Table 4, we analyze how different loss function designs influence model performance when trained on our constructed preference data. As shown in the second line of the table, using a naive DPO loss paired with our self-rewarded preference data could achieves consistent improvements on LLaVA-OV-7B, which serves as a strong baseline. Then, by equipping with length normalization, the DPO loss could avoid the effect of length differences between the preferred and dispreferred preference data, thus improving performance on several benchmarks. It also effectively control the increase of output length brought be the training. We then evaluate the design visual weighting strategies. We ablate our designed token-level visual weight by varying the scaling factor $\alpha$ in Equation 13. Our default setting ($\alpha = 0.5$) achieves the best balance across general benchmarks. These results highlight that our modified loss design more effectively improve the image preference of MLLMs.

Table 4: Experiments on loss function designs. "Length Norm" denotes the proposed length-normalized loss. "Length" means the averaged response length on 500 image description instructions from the RLAIF dataset.

| Model | Seed2Plus | OCR | MMMUPro | LLaVA | MMStar | MMB | Length |
|---|---|---|---|---|---|---|---|
| LLaVA-OV-7B | 64.8 | 62.1 | 29.8 | 90.7 | 61.7 | 80.8 | 164.4 |
| + DPO | 66.3 | 67.4 | 30.5 | 93.1 | 62.3 | 80.8 | 180.3 |
| + Length Norm (ours) | 66.6 | 68.4 | 30.5 | 92.8 | 62.7 | 80.8 | 170.9 |
| + Token Visual Weight ($\alpha$ in Equation 13) | | | | | | | |
| $\alpha = 0.4$ | 66.4 | 67.7 | 30.4 | 93.0 | 62.7 | 81.0 | - |
| $\alpha = 0.5$ (default) | 66.6 | 67.9 | 30.6 | 93.2 | 62.8 | 81.2 | - |
| $\alpha = 0.6$ | 66.6 | 67.9 | 30.5 | 93.1 | 62.6 | 80.8 | - |

Table 5: Experiments on previous preference datasets on LLaVA-OV-7B. All models are fine-tuned under the same training configuration.

| Model | Seed2Plus | OCRBench | MMMUPro | MMStar | MMB | POPE (F1) | MMHal |
|---|---|---|---|---|---|---|---|
| LLaVA-OV-7B | 64.8 | 62.1 | 29.8 | 61.7 | 80.8 | 88.1 | 2.97 |
| + RLHF-V (Yu et al., 2024a) | 65.2 | 62.3 | 29.6 | 62.1 | 80.8 | 88.1 | 2.91 |
| + RLAIF-V (Yu et al., 2024b) | 64.8 | 62.5 | 29.6 | 62.2 | 80.8 | 88.5 | 2.86 |
| + POVID (Zhou et al., 2024a) | 65.1 | 65.2 | 29.5 | 62.0 | 81.1 | 89.6 | 2.99 |
| + ConSR (Ours) | 66.6 | 67.9 | 30.6 | 62.8 | 81.2 | 88.7 | 3.30 |

## 4.4 COMPARISON WITH OTHER PREFERENCE DATASETS.

Table 5 compares our proposed method, ConSR, with several existing preference alignment methods, including RLHF-V(Yu et al., 2024a), RLAIF-V(Yu et al., 2024b), and POVID (Zhou et al., 2024a). To avoid the effect of data scale, we sample up to 10K preference pairs from each method and fine-tune LLaVA-OV-7B using the same training configurations. These methods construct preference data using diverse sources such as SFT ground-truth data, MLLM-generated responses, or corrupted ground-truth data, with supervision derived from human annotators, GPT-4V, or other external MLLMs. Across all benchmarks, ConSR outperforms prior methods on general image understanding tasks and achieves competitive performance on hallucination benchmarks. Notably, ConSR shows the largest improvements on MMMU-Pro (+0.8) and SeedBench2Plus (+1.8), demonstrating its effectiveness in enhancing both multimodal reasoning and OCR-related capabilities. While POVID performs slightly better on POPE, ConSR surpasses it on MMHal (+0.31) and provides more consistent gains across all evaluated benchmarks. These results highlight ConSR as a generalizable alternative to existing preference alignment methods, achieving strong performance without reliance on external supervision.

## 5 CONCLUSION

In this work, we propose Contrastive Self-Rewarding approach (ConSR), a self-supervised framework for aligning multi-modal large language models (MLLMs) without relying on human annotations or external reward models. By leveraging contrastive input perturbations, including both corrupting fine-grained visual details or enriching semantic context, we derive self-reward signals that reflect the model's sensitivity to input visual information. These signals are used to construct preference data that well reflects the consistency towards the input visual context. We further enhance preference optimization by reformulating the DPO objective with length normalization and token-level visual weighting. Extensive experiments on a wide range of image understanding and hallucination benchmarks demonstrate that ConSR consistently improves performance across different model architectures, highlighting its effectiveness and generality. Our work offers a scalable, annotation-free alternative for preference alignment, paving the way for more trustworthy MLLMs with comprehensive multimodal capabilities.

**Ethics statement.** This work adheres to ICLR's ethics requirement. The study does not involve any human subjects. The involved datasets are contructed based on publicly available data and will be open-sourced. We have taken care to avoid possible harmful insight, discrimination/bias/fairness concerns, privacy and security issues, legal compliance, or research integrity issues.

**Reproducibility statement.** We have illustrated all required details to reproduce the experiments, including implementation details, model configurations, the process of constructing data. We believe these measures and details will enable other researchers to reproduce our work and further explore the field.

**LLM Usage.** In this work, the LLM is used in two aspects. (1) Paper writing: we adopt GPT to help write and polish the manuscript, such as refining the language to increase readability, finding possible typos. (2) Evaluation of data quality: in the appendix, we present a comparison between the preference data constructed by our proposed ConSR and the rankings generated by GPT for the same data, thereby demonstrating the effectiveness of the preference data.

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

## A  APPENDIX

## B  IMPLEMENTATION DETAILS

### B.1  PREFERENCE DATA CONSTRUCTION.

For sampling the responses of MLLMs, we set the temperature as 1.2 and sampling 5 outputs for each prompts in the datasets. For noisy sampling, we apply 500 steps of diffusion Gaussian noise on the original image and also sample 5 outputs with the temperature of 1.2. To revise responses, we first apply the following prompts for MLLMs to generate image descriptions:

> Describe the given image in detail. Compress the description in one paragraph. Please inlucde all foreground and background contents such object types, colors, states, actions, number of objects, precise object locations, texts or OCR results, relationships and relative positions between objects, environment, background, time, weather, etc. In case of non-real-world scenes, like charts, graphs, tables, etc., please describe the table, mention all numbers (if any), mention the written text, and all other details.

Then, we use the following prompts for MLLMs to revise their original responses while remain the original structure and language style:

> You are given an image, its description, a question or an instruction about the image, and your original response. Carefully review all information, and revise your original response only if necessary to make it more accurate and clearly grounded in the image or description. Preserve the original phrasing and structure as much as possible. If the original response is already correct, leave it unchanged. Do not invent details not supported by the image. Provide the final revised response without any prefix directly.
> Description:
> Question:
> Original Response:

As illustrated in Section 3.2, we leverage two types of modifications on the inputs for contrastive rewarding. We apply 800 steps of diffusion Gaussian noise on the input images to corrupt the input visual details. To enhance input semantics, we supplement the input prompt with image descriptions (generated by MLLMs themselves based on the above prompt) as follows:

> For your reference, the input image can be described as follows:
> {Image Description}
> {Original Prompt}

### B.2  TRAINING SETTINGS.

For training, we adopt LORA with the rank of 128. We set the learning rate to 2e-7 with a batch size of 1 and 2 for LLaVA-OneVision and Qwen2.5-VL, respectively. All the models are trained for 1 epoch. For LLaVA-OneVision, we adopt their official implementation on DPO training, while LLaMA-Factory Zheng et al. (2024) is adopted for the training of Qwen2.5-VL-7B.

## C  MORE ABLATIONS

**Noise step for contrastive rewarding.** We experiment with the noise step for contrastive input as illustrated in Section 3.2. As shown in Table 6, a moderate noise level (low diffusion steps) has little effect and may even lead to performance degradation. This can be attributed to the fact that most fine-grained image details are still preserved, resulting in insufficient contrast in output probabilities to distinguish between MLLM responses effectively. As the number of noise steps increases, the model exhibits consistent improvements on both the hallucination benchmark, POPE, and general

Table 6: Noise Steps for corrupting visual details in contrastive rewarding. All the experiments are conducted with normally sampled responses and original DPO loss.

| Model | POPE | MMStar | MMB | MMMU_Pro | MME |
|---|---|---|---|---|---|
| LLaVA-OV-7B | 88.1 | 61.7 | 80.8 | 29.8 | 1998 |
| noise step: 200 | 87.9 | 62.0 | 81.0 | 29.2 | 1988 |
| noise step: 400 | 88.0 | 62.2 | 81.0 | 29.5 | 1993 |
| noise step: 600 | 88.0 | 62.7 | 81.2 | 29.3 | 1992 |
| noise step: 800 | 88.6 | 63.0 | 81.4 | 30.1 | 1998 |
| noise step: 999 | 88.6 | 62.8 | 81.2 | 29.9 | 1986 |

image understanding tasks. Based on this observation, we adopt 800 diffusion steps as the default setting in our experiments.

**Other methods for Visual Corruption.** As shown in Table 7, we experiment with a broader set of visual corruption methods when computing detailed scores, including random masking, resize-cropping, mixup, and saliency-based background masking in place of diffusion noise. The diffusion noise achieves slightly better performance compared with other corruption methods.

Table 7: Experiments of different visual corruption methods.

| Method | POPE | MMB | MMStar | OCR | Seed2Plus |
|---|---|---|---|---|---|
| noise (default) | 86.5 | 84.1 | 65.5 | 88.6 | 71.1 |
| random mask | 86.4 | 83.8 | 63.6 | 88.6 | 71.1 |
| resize crop | 86.3 | 83.6 | 63.7 | 88.4 | 71.0 |
| mix up | 86.2 | 84.3 | 63.8 | 88.4 | 70.8 |
| salient masking background | 86.3 | 83.8 | 63.3 | 88.3 | 70.9 |

**Comparison of different texts for enhancing input semantics.** As shown in Table 8, we ablate different texts for enhancing input semantics. The Qwen2.5-VL-7B is adopted as the base model. Three texts are compared including image descriptions from model with different sizes, Qwen2.5-VL-7B and Qwen2.5-VL-32B, as well as an enhanced question answering prompt: "Before answering, ensure your response is directly grounded in the visual content, not overly influenced by general world knowledge or unsupported assumptions. Avoid speculation, hallucinated facts, or vague language. Responses must reflect a high degree of visual precision and stay tightly aligned with the observable evidence, maintaining visual consistency at all times.". The image description from larger model achieves similar performance. However, the enhanced only QA prompt is much lower than image description. We conjecture that it is due to the contrast probability shift from image description could offer more supervisions to the input image semantic.

Table 8: Experiments of different texts for enhancing input semantics.

| Model | POPE | MMB | MMStar | OCR | Seed2Plus |
|---|---|---|---|---|---|
| Description from Qwen2.5-VL-7B | 86.5 | 84.1 | 65.5 | 88.6 | 71.1 |
| Description from Qwen2.5-VL-32B | 86.4 | 83.9 | 65.3 | 88.9 | 71.0 |
| Enhanced Question Answering Prompt | 86.5 | 83.7 | 64.6 | 88.3 | 71.0 |

**Upper and lower bounds of the token visual weight ($\epsilon$ in Equation 12).** In Section 3.3, we introduce a parameter $\epsilon$ to constrain the upper and lower bounds of the token-level visual weight, preventing certain tokens from dominating the loss. We evaluate different clipping ranges in Table 9 and observe that the performance is relatively insensitive to varying different $\epsilon$. Based on this, we set $\epsilon = 0.2$ as the default value in our experiments.

**Effect of Length-normalization on response length.** In Table 4, we demonstrate that our proposed length normalization, by alleviating the length bias of preference data in the DPO loss, effectively improve MLLMs image understanding capabilities. To further evaluate the impact of length normalization, we examine its effect on the length of model responses. Specifically, we sample 500 image

Table 9: Upper and lower bounds for token-level visual weight ($\epsilon$ in Equation 12).

| Model | MMStar | MMB | MMMU_Pro | Seed2Plus | LLaVA |
|---|---|---|---|---|---|
| $\epsilon = (-0.2, 0.2)$ | 62.4 | 81.2 | 30.4 | 66.5 | 93.1 |
| $\epsilon = (-0.1, 0.2)$ | 62.4 | 81.0 | 31.0 | 66.6 | 91.7 |
| $\epsilon = (0, 0.1)$ | 62.4 | 80.8 | 30.4 | 66.7 | 91.5 |
| $\epsilon = (0, 0.2)$ | 62.8 | 81.2 | 30.6 | 66.6 | 93.2 |
| $\epsilon = (0, 0.3)$ | 62.9 | 81.0 | 30.6 | 66.6 | 93.1 |

description instructions from the RLAIF dataset and compare the average response length with and without length normalization. Without normalization, the average response length is 180.3 words, whereas applying our length normalization reduces it to 170.9 words. This result confirms that our method effectively shortens the model's responses with the same preference data, indicating better control over verbosity without compromising alignment quality.

**Comparison with other DPO data on Qwen2.5-VL-7B.** As shown in Table 10, we fine-tune Qwen2.5-VL-7B with various previous DPO datasets, including RLHF-V, RLAIF-V, and POVID. These datasets could effectively alleviate hallucination, yielding much higher POPE scores, which match their original intuition for hallucination mitigation. However, their gains on general-purpose and reasoning benchmarks remain limited. In contrast, ConSR consistently outperforms all baselines across both hallucination and complex multimodal reasoning tasks, with improvements exceeding +1% on several benchmarks. We attribute this to the contrastive reward's ability to guide the model toward stronger alignment with fine-grained visual input, benefiting not only factual grounding but also downstream visual reasoning.

Table 10: Experiments on previous preference datasets on Qwen2.5-VL-7B.

| Model | POPE | MMB | MMStar | OCR | Seed2Plus | MathVista |
|---|---|---|---|---|---|---|
| Qwen2.5-VL-7B | 87.3 | 83.7 | 64.4 | 88.2 | 70.6 | 68.2 |
| POVID | 88.2 | 84.5 | 63.6 | 88.0 | 70.9 | 67.6 |
| RLHF-V | 85.6 | 83.9 | 65.6 | 88.5 | 70.4 | 67.2 |
| RLAIF-V | 87.1 | 84.1 | 64.6 | 88.5 | 71.2 | 68.5 |
| ConST (ours) | 86.4 | 84.1 | 65.5 | 88.6 | 71.1 | 69.8 |

# D PREFERENCE DATA QUALITY

**Consistency of Preference compared to GPT and Human Annotators.** To further address the remaining points, we conducted additional analyses on the quality of our response rankings by comparing those produced by our contrastive self-reward (ConSR) framework with those from human annotators and GPT-4o-mini. The comparison includes both pairwise and ranking-wise evaluations.

For pairwise comparison, We sampled 400 pairs of accepted vs. rejected responses from our DPO training data and asked two human experts to independently judge which response in each pair was better. Additionally, we prompted GPT to evaluate the same pairs. We then compared both ConSR's preferences and GPT's preferences against the human annotations.

- ConSR–human agreement: 81% of ConSR-labeled preferences align with human judgments.
- GPT–human agreement: 77% of GPT-labeled preferences align with human judgments.

These results suggest that ConSR-generated preferences are highly consistent with human evaluations and comparable in quality to those produced by strong AI models.

To assess overall ranking quality, we selected 500 examples and asked GPT to rank the 15 responses of each example. We then computed the Spearman correlation between GPT's rankings and the rankings derived from ConSR scores.

- Median Spearman correlation: 0.448

- 70th percentile of Spearman correlation: 0.662
- Preserved chosen-rejected orderings: 79.4% (the ratio of where best and worst responses identified by ConSR maintained the same relative order in GPT's rankings.)

These results demonstrate that the contrastive reward induced by input perturbations serves as an effective signal for response quality—yielding rankings broadly aligned with both human and GPT-4o evaluations. We observe that the top-ranked responses selected by ConSR typically capture more visual details and semantic information. These content tend to be more sensitive to changes in the visual input and likely contribute to improved response quality.

