# OpenReview forum: "Contrastive Self-Rewarding MLLM"
_ICLR.cc/2026/Conference — ICLR 2026 Conference Withdrawn Submission_

### Official Review · Reviewer_zhtt · 2025-10-29

**Soundness:** 3
**Presentation:** 3
**Contribution:** 3
**Rating:** 6
**Confidence:** 3

**Summary:**

This paper proposes ConSR, a way to align multimodal large language models without using human labels or external reward models. The main idea is to measure how much a model’s answer changes when the input image is either degraded or enriched. If the model reacts in a meaningful way to those changes, that behavior becomes a “self-reward” signal.
ConSR builds preference pairs from those signals and fine-tunes the model with a modified DPO loss that (1) normalizes for response length and (2) gives more weight to visually sensitive tokens.
Experiments on LLaVA-OneVision and Qwen2.5-VL show clear and consistent improvements on both general multimodal benchmarks and hallucination benchmarks.

**Strengths:**

1) The idea of using the model’s own probability shifts under contrastive inputs as a reward is creative and well motivated. It avoids external judges and keeps everything self-contained.
2) The improvements are small but steady across tasks, which makes the method look robust.
3) The experiments are thorough: multiple backbones, diverse benchmarks, plus solid ablations

**Weaknesses:**

1) The novelty is more in the combination than in each part. Self-rewarding and contrastive cues have both been seen before, so the real contribution lies in putting them together neatly.
2) Most gains are within 1–2 points, which are meaningful but modest; it would help to show some qualitative or human evaluation to make the improvement more tangible.
3) A small discussion about where the approach might break would strengthen the paper.

**Questions:**

1)  How large is the overlap between the self-rewarded pairs and those that GPT-4 would consider preferred?
2) Have you checked whether the method encourages shorter or longer answers beyond what length normalization controls?

---

### Official Review · Reviewer_4o6s · 2025-10-31

**Soundness:** 2
**Presentation:** 3
**Contribution:** 2
**Rating:** 2
**Confidence:** 5

**Summary:**

The authors propose a framework named ConSR (Contrastive Self-rewarded Preference Optimization) to address a key limitation in aligning Multimodal Large Language Models (MLLMs): the reliance of Direct Preference Optimization (DPO) on human-annotated or model-based preference datasets.

The core idea is to create a "self-rewarding" mechanism that bypasses the need for external supervision. The framework operates by:

- **Constructing Contrastive Inputs**: Automatically perturbing the visual input by (1) degrading fine-grained details (e.g., blurring) and (2) enriching semantic context (e.g., adding descriptions).

- **Generating a Self-Reward Signal**: The MLLM generates responses for both the original and perturbed inputs. The core hypothesis is that the magnitude of change in the model's response can serve as a reward signal.

- **Constructing Preference Pairs**: This reward signal is used to automatically rank different model outputs, thereby algorithmically constructing preference pairs (e.g., "this response > that response").

- **Optimization**: These auto-generated preference pairs are then used to fine-tune the MLLM using DPO. The authors also introduce modifications to the DPO objective to mitigate length bias and re-weight visual tokens.

**Strengths:**

1. **Valid Motivation**: The paper tackles a significant and well-recognized problem in the field: the high cost and scalability bottleneck of acquiring high-quality preference data for alignment. The goal of creating an unsupervised preference alignment framework is well-motivated.

2. **Technical Additions**: The authors demonstrate an understanding of DPO's limitations by including specific technical modifications, such as the adjustment for length bias and the re-weighting of visual tokens, which go beyond the high-level concept.

**Weaknesses:**

This paper's foundational premise suffers from severe flaws, and its core methodology is questionable.

1. **Fundamental Flaws of the Proxy Reward**: This is the most critical weakness. The paper's central hypothesis—that a model's sensitivity to visual perturbation is a valid proxy for human preference—***is a highly problematic and unsubstantiated leap of faith***.

  - **Misalignment with Human Preference**: The authors provide no compelling evidence that this proxy metric (sensitivity) correlates with what humans actually value (e.g., factual accuracy, logical coherence, safety, or helpfulness). At best, this signal is **insufficiently aligned with human preference**.

  - **Obvious "Hackability"**: This proxy reward is **transparently "hackable"**. A model could easily learn to "game" this signal by "performing" sensitivity—for example, by strategically outputting high-variance responses to any perceived perturbation—without making any genuine improvements to its core capabilities. The paper **completely fails to discuss or investigate this critical risk of reward hacking**.

  - **Underexplored "Black Box" Signal**: The paper uses perturbation sensitivity as a reward, but **what this signal is *actually* rewarding remains entirely underexplored**. It is a "reductonist" approach, simplifying the complex, multi-dimensional nature of human preference into a single, simplistic metric of "change."

  - **"Echo Chamber" and Ungrounded Drift**: Using this internal, unexplored signal as a reward creates a classic **"echo chamber"**. Without any external, grounded truth as an anchor, the model is guided solely by its own internal artifacts. This will inevitably lead the model to amplify its own biases and flaws, causing it to "drift" from factual or useful grounding.

  - **Self-Contradictory Evaluation**: The paper ultimately relies on expensive human evaluation to prove its method's efficacy. This is a **fundamental self-contradiction**: it demonstrates that the authors themselves cannot trust this "self-rewarding" signal to be a reliable substitute for human judgment, thereby exposing the signal's inherent unreliability.

2. **Significant Lack of Novelty**: The core mechanism—using the difference in responses or logits between original and perturbed inputs as a reward signal—**is not novel**. This concept has been explored in prior work within the multimodal domain (e.g., POVID [1], VCD [2], SeVA [3], and more recently, VPPO [4], [5]). The paper's contribution appears to be repackaging this known technique and applying it to DPO, which constitutes a very limited methodological advance.

3. **Outdated and Ineffective Baselines**: The paper's primary baseline is DPO. This is an increasingly outdated choice, as the field is rapidly moving towards online and on-policy alignment methods. Comparing against a baseline that no longer represents the state-of-the-art makes the claimed "improvements" far less compelling.

4. **Insignificant and Statistically Questionable Improvements**: Many of the reported performance gains are marginal. They could very likely be attributed to statistical noise or normal evaluation variance. The authors fail to provide any statistical significance analysis, leaving the reader to conclude that the method may not offer any tangible, generalizable benefits.



[1] Aligning Modalities in Vision Large Language Models via Preference Fine-tuning

[2] Mitigating Object Hallucinations in Large Vision-Language Models through Visual Contrastive Decoding

[3] Self-Supervised Visual Preference Alignment

[4] SPOTLIGHT ON TOKEN PERCEPTION FOR MULTIMODAL REINFORCEMENT LEARNING

[5] On Epistemic Uncertainty of Visual Tokens for Object Hallucinations in Large Vision-Language Models

**Questions:**

1. **On Reward Hacking**: The proposed proxy reward seems trivially "hackable." Did you conduct any experiments to test for this? For example, does the model learn to simply maximize response variance upon detecting any input perturbation, regardless of its core performance on the task? How can you guarantee this signal is not being exploited?

2. **On Reward Justification**: Your core assumption is that perturbation sensitivity correlates with human preference. What evidence supports this beyond the final, indirect benchmark scores? What, exactly, do you believe this reward signal is incentivizing the model to learn?

3. **On Novelty**: The mechanism of using perturbation-based response diffs as a signal has been seen in prior work (e.g., VCD, SeVA). Could you please precisely articulate the novel methodological contribution of ConSR, distinguishing it from these previous applications of a similar concept?

4. **On Statistical Significance**: Many of your reported gains are marginal. Have you performed statistical significance tests (e.g., bootstrap or permutation tests) to confirm that these improvements are real and not simply evaluation noise?

5. **On Baselines**: Why did you choose to compare against DPO, an increasingly outdated baseline, rather than more current state-of-the-art online or on-policy alignment methods?

---

### Official Review · Reviewer_A4LB · 2025-11-01

**Soundness:** 2
**Presentation:** 2
**Contribution:** 2
**Rating:** 2
**Confidence:** 5

**Summary:**

This paper explores the problem of aligning multimodal large language models (MLLMs) without human preference data or external reward models. The key contribution is the Contrastive Self-Rewarding framework (ConSR) with visual perturbations to construct contrastive visual pairs, using differences in the model’s responses as self-supervised reward signals. A modified DPO objective with length normalization and visual-token reweighting is used to improve training stability and multimodal grounding. The paper is clearly written and experimentally evaluated on standard multimodal benchmarks.

**Strengths:**

- This paper has good motivation to study an important problem.
- Introducing visual perturbation as a contrastive mechanism is a creative and intuitive extension of self-rewarding LLM frameworks.
- The paper is clear, well-structured, and easy to follow, with a good set of benchmark coverage.

**Weaknesses:**

- The paper explores only a limited set of visual editing methods (diffusion-based degradation). More perturbation types or systematic analysis are needed to justify that the method generalizes beyond specific image transformations. Generative methods could be explored to see visual variety and complexity changes in perturbation. Following from the previous point, there are no visual examples or detailed explanations showing how perturbations influence model outputs or why such perturbations help the model learn richer visual semantics.

- The claimed “self-rewarding” property may not generalize beyond these carefully controlled perturbations; it is unclear whether the approach would remain effective on natural or unseen image variations.

- The experiments in the main body of the manuscript only include comparisons with vanilla models as baselines. In Table 10, the authors further present results against other existing methods; however, the proposed approach does not demonstrate significant improvements over these baselines.

- The paper would benefit from a clearer problem formulation and a more precise explanation of which dimensions of VLM performance the proposed approach targets.

- The general framework has been extensively explored in prior work on VLM alignment. From my perspective, this paper mainly applies existing techniques and their minor extension, without introducing substantially new ideas.

- The presentation quality needs improvement to meet academic standards. For example, the paper incorrectly refers to 'probability shift'—the correct concept is 'distribution shift'.

**Questions:**

See Weaknesses

---

### Official Review · Reviewer_Pkf9 · 2025-11-06

**Soundness:** 3
**Presentation:** 3
**Contribution:** 3
**Rating:** 6
**Confidence:** 3

**Summary:**

The paper presents ConSR, a contrastive self-rewarding framework for preference optimization in Multi-modal Large Language Models (MLLMs). Instead of relying on external reward models or human annotations, ConSR perturbs visual inputs (via degradation or semantic enhancement) to construct contrastive pairs and derives self-reward signals from model output sensitivity. A length-normalized DPO loss with token-level visual weighting mitigates length bias and enhances fine-grained visual grounding. Experiments on multiple benchmarks show consistent improvements over strong baselines such as LLaVA-OneVision and Qwen2.5-VL.

**Strengths:**

1. Innovative Self-supervised Alignment: Introduces a creative, annotation-free preference optimization approach, significantly reducing dependency on human or external rewards.
2. Sound Theoretical Design: The reformulated DPO loss effectively addresses known weaknesses of traditional DPO. Mathematical exposition is clear and reproducible.
3. Comprehensive Experiments: Evaluations across 11 benchmarks with thorough ablations substantiate all key claims.

**Weaknesses:**

1. Baseline Comparisons Outdated: Lacks direct comparison with the newest self-rewarding or contrastive alignment methods, such as CHiP, AdaViP, MIA-DPO, and MCM-DPO.
2. Insufficient Analysis of Perturbation Design: Diffusion-based perturbation is empirically best but theoretically underexplained.


[1] CHiP: Cross-modal Hierarchical Direct Preference Optimization for Multimodal LLMs
[2] AdaViP: Aligning Multi-modal LLMs via Adaptive Vision-enhanced Preference Optimization
[3] MIA-DPO: Multi-Image Augmented Direct Preference Optimization For Large Vision-Language Models
[4] MCM-DPO: Multifaceted Cross-Modal Direct Preference Optimization for Alt-text Generation

**Questions:**

1. How does ConSR scale to architectures with weaker visual grounding or video inputs?
2. Can the self-reward mechanism lead to pathological or unstable learning behavior?

---

### Note · Authors · 2025-11-14

I have read and agree with the venue's withdrawal policy on behalf of myself and my co-authors.